# Study on the Synergistic Evolutionary Effects of China's Digital Economy Core Industry and Energy Industry Based on DEA Malmquist Synergistic Development Model and Grey Correlation Analysis

Guoteng Xu [1] 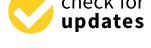, Jingwei Zhu [1,*], Chengjiang Li [2] and Jingtong Shan [3]

1   State Key Laboratory of Public Big Data, Guizhou University, Guiyang 550025, China; gtxu@gzu.edu.cn
2   School of Management, Guizhou University, Guiyang 550025, China; chengjiang.li@utas.edu.au
3   School of Public Policy and Urban Affairs, Northeastern University, Boston, MA 02118, USA; shan.ji@northeastern.edu
*   Correspondence: gs.jwzhu22@gzu.edu.cn

**Abstract:** The burgeoning digital economy has facilitated a transformation and upgraded within the energy industry, which, in return, continually guarantees robust energy security for the expansion of the digital economy. China's digital economy and energy sector have increasingly merged and innovated in the domains of technology, market, and operations in recent years. Consequently, an accurate assessment of the interplay between these two sectors and their evolving patterns is vital for policy formulation and execution concerning their joint development. Drawing on 14,520 authoritative departmental statistics from 30 Chinese provinces spanning 2011 to 2021, this study applies techniques such as Data Envelopment Analysis (DEA)-Malmquist, grey correlation, and objective empowerment to develop a quantitative evaluation model for the reciprocal evolution of these industries finding that the own synergistic evolutionary effect of these two industries experienced fluctuations, declining from 0.8512 and 0.7535 in 2012 to 0.4590 and 0.4378 in 2021, respectively. Conversely, the comprehensive synergistic evolutionary effect between industries increased from 0.5879 in 2012 to 0.6841 in 2021. Building upon these findings, a series of development proposals are put forth to provide valuable insights and recommendations for advancing the high-level coordinated development of China's digital economy and energy industry.

**Keywords:** collaborative evolution of digital economy and energy industry; DEA Malmquist model; grey correlation analysis; coupling coordination; objective weighting method



## 1. Introduction

Recognized as a significant driver of global economic progression, the digital economy hinges on the pivotal role of the energy sector for its growth. According to the "2022 Global Digital Economy White Paper" released by the China Academy of Information and Communications Technology (CAICT), China's digital economy has made remarkable strides and is now second only to the United States in scale [1]. Amid China's economic transition marked by the "three-phase overlap" of growth transition, structural adjustment, and the accumulation of previous stimulus policies, novel paradigms and business models steered by the digital economy are poised to foster high-quality economic development. Research from the World Economic Forum indicates that a 10% surge in digitization corresponds to a 0.5% to 0.62% increase in per capita GDP growth. Consequently, collaborative growth between the digital and physical economies could expedite the transformation of China's economic growth engines. This transition is vital for establishing a modern industrial system, achieving superior-quality development, and forging a new development pattern.

Energy is an indispensable resource propelling economic and societal development, forming the bedrock of national economic progress. As the global leader in energy consumption and production, China's energy sector is faced with the imperative of undergoing five transformative shifts. These include transitioning from high- to low-carbon pathways, replacing traditional fossil fuels with renewable energy sources, upgrading from conventional to smart energy technologies, shifting from governmental regulation to market guidance, and transitioning from a domestic to an international supply system. Notably, numerous provinces in China's western regions—a focal point for energy sector development—experience substantial mismatches in their energy components and corresponding industries. Moreover, despite being a country abundant in coal and deficient in oil, China officially pledged at the 75th United Nations General Assembly to peak carbon dioxide emissions by 2030 and attain carbon neutrality by 2060. Under the acute pressure to restructure the energy sector, elevate industry standards, and curtail carbon emissions, China's digital economy presents an opportune solution. The latter can markedly reduce carbon emissions by improving energy efficiency and restructuring energy systems through the proliferation of the digital economy. Hence, examining the collaborative growth of the digital economy and the energy sector holds practical significance and theoretical value in promoting high-quality economic development in China.

The digital economy and the energy industry, two critical pillars of China's economic development, demonstrate autonomous and interdependent characteristics. The burgeoning digital economy imposes fresh demands on the energy sector, including substantial electricity needs for large data centers and cloud computing infrastructure. Conversely, the evolution of the energy industry introduces novel prerequisites for the digital economy, such as efficient energy utilization, clean production, and the application of information technology to develop integrated service platforms based on big data, cloud computing, and the Internet of Things. These innovations necessitate digital technology support. Given the brisk development of China's digital economy and energy industry in recent years, the focal point of this article is the internal and external synergies of these industries and their relation to industry growth.

This study addresses the following scientific inquiries: (1) What characterizes the correlation between the internal synergies of China's core digital economy industry and energy industry and their evolution? (2) How do the external synergies of these industries relate to their progression? By addressing these queries, this article intends to elucidate the role of the synergistic evolutionary impacts of China's core digital economy industry and energy industry in facilitating the conversion process from traditional to emerging kinetic energy, thereby promoting high-quality industrial development.

This article is structured into five sections. Section 1 serves as the introduction, providing the research background, scientific questions, and outlining the marginal contributions of this paper. Section 2 presents a comprehensive literature review, covering the progress made in the domestic and foreign research on the "Digital Economy Core Industry", the "Energy Industry", and the synergistic development between them. Section 3 focuses on the research process, index system, and model construction. It details the research progress, constructs an index system consisting of 13 indicators across four dimensions for both the Digital Economy Core Industry and the Energy Industry. The research process is divided into four parts to investigate the industry's own synergistic evolutionary effect, the inter-industry synergistic evolutionary effect, the effective synergistic evolutionary effect of comprehensive performance between industries, and the inter-industry synergistic evolutionary effect. Section 4 comprises data indicators, empirical analysis, and discussion of results. The data utilized in this study are obtained from inter-provincial panel data spanning from 2011 to 2021, focusing on 30 provinces in China. The indicators are selected based on the latest industry classification authority document, "National Economic Classification of Industries" (GB/T4754-2017). In the empirical analysis section, based on the model constructed before, the article conducts empirical analysis on the industry's own synergistic evolutionary effect, the inter-industry synergistic evolutionary effect, the effective



synergistic evolutionary effect of comprehensive performance between industries, and the inter-industry synergistic evolutionary effect, exploring the relationships between internal and external synergistic effects of the Digital Economy Core Industries and the Energy Industry and their impact on industrial development. The results of the empirical analysis are systematically summarized and discussed from the internal and external. Finally, in Section 5, policy recommendations are presented for the Digital Economy Core Industry, the Energy Industry, and the synergistic development between them. Additionally, possible directions for future research are also presented.

This study makes a marginal contribution in the following ways: (1) while current research on the comprehensive synergistic evolution effects between distinct systems is limited, particularly regarding the integration of effective synergistic effects of comprehensive performance as part of the system, this study broadens the scope of research through theoretical exploration and empirical analysis; (2) utilizing the DEA synergy development model and grey relational analysis and considering the timeliness and availability of indicators, this study examines 11-year panel data from 30 provinces in China. This comprehensive investigation illuminates the synergistic evolution effects within and outside China's Digital Economy Core Industry and Energy Industry, providing novel perspectives and insights into the relationship between industrial synergistic evolution effects and industrial development.

## 2. Literature Review

### 2.1. Digital Economy Core Industry

In the Digital Economy Core Industry field, authoritative institutions and scholars have conducted extensive research. The International Telecommunication Union (ITU) constructed the ICT Development Index (IDI) from three aspects: ICT access, ICT usage, and ICT skill level [2]. The Bureau of Economic Analysis (BEA) of the United States divides the digital economy into three categories—digital infrastructure, e-commerce, and digital media—from the supply-side perspective combined with the characteristics of ICT technology [3]. The European Union constructed the Digital Economy and Society Index (DESI) from five dimensions: connectivity, digital skills, internet application, integration of digital technology, and digital public services [4]. The CAICT constructed the Digital Economy Index from three dimensions: advance, consistency, and lagging, considering digital industrialization, industrial digitization, digital governance, and the impact of the digital economy on macroeconomic development [5]. Bruno et al. focused on the Digital Economy and Society Index (DESI) and used correlation and principal component analysis to study the data from 29 E.U. countries in 2020, evaluating the digital divide issues between and within countries [6]. Imran et al. studied the direct impact of DESI dimensions on Sustainable Goal Development Index (SGDI) using a panel regression model and found that connectivity, human capital, and internet service usage have a greater negative impact on SGDI than digital technology and digital public services [7]. Chauhan et al. found through empirical research that the Internet of Things and Artificial Intelligence play a critical role in the transition to Circular Economy (CE) [8]. D. U. et al. used the D-G coefficient decomposition method and the panel spatial econometric model to analyze the coupling and coordinated development level between China's digital economy and rural revitalization and found that the coupling and coordination between the two have gradually improved [9]. Wang et al. explored the impact and mechanism of digital technology innovation and technology spillovers on domestic carbon intensity based on Organization for Economic Co-operation and Development (OECD) data, the Koopman, Powers, Wang and Wei (KPWW) method, and multiple panel regressions [10]. Zhou et al. quantitatively evaluated the impact of broadband infrastructure development on China's urban export trade and found that broadband infrastructure can significantly promote the growth of urban export trade [11]. Ma et al. measured the level of China's urban digital economy and high-quality green development and found that the digital economy has a nonlinear positive effect on high-quality green development [12]. Wang et al. used

machine learning algorithms to establish a quality evaluation model for the development of digital economy in Shaanxi Province and accurately measured the level of digital economy development in Shaanxi Province using authoritative data [13].

In summary, previous studies on digital economy measurement mainly adopt a "stratified classification" approach, and there are certain differences among various indicator systems. This article will combine China's national economic industry classification method and integrate industries such as telecommunications equipment and services, computer software, and computer hardware into the scope of the Digital Economy Core Industry, providing a scientific basis and authoritative data resource guarantee for subsequent research.

### 2.2. Energy Industry

Researchers in related fields have carried out a series of productive studies. Shahzad et al. found through empirical analysis that economic growth and urbanization have increased energy consumption [14]. Acheampong et al. found that economic, social, and political globalization have an inverted U-shaped relationship with economic growth, and economic, social, and political globalization and energy consumption also have a U-shaped relationship. [15]. Li et al. proposed other influential indicators for measuring energy efficiency performance based on a summary of industrial energy efficiency evaluation index categories [16]. Rafiq et al. found that population density and affluence have increased emissions and energy intensity, while the level of openness has significantly reduced pollutant emissions and energy intensity while urbanization has significantly increased energy intensity [17]. Mohsin et al. found a positive correlation between economic growth and energy consumption and that a 1% increase in renewable energy consumption reduces carbon emissions by 0.193% [18]. Chen et al. analyzed the Middle East and North African countries' authoritative statistical data from the perspective of sustainable development and found that technological innovation and economic structural transformation have a positive impact on energy efficiency [19]. Mughal et al. found a bi-directional causal relationship between economic growth and energy use [20]. Jiang et al. used the input–output method, structural decomposition method, and energy use method to investigate the structural emission reduction of China's electric power heating industry and found that low-carbon energy sources such as coke oven gas, converter gas, blast furnace gas, and natural gas generally have emission reduction effects [21]. Ehsanullah et al. measured and analyzed the energy, economic, social, and environmental performance of the Group of Seven (G7) countries and their energy poverty index using the data envelopment analysis (DEA) method [22]. Muhammad et al. used the slack-based measure (SBM) DEA model and random effects (RE) and fixed effects (F.E.) models to study the nonlinear relationship between the industrial structure, energy intensity, and environmental efficiency of developed and developing countries and found that the average environmental efficiency of developed countries is higher than that of developing countries [23].

In summary, previous studies have conducted an in-depth exploration and analysis of issues related to the development of the Energy Industry, energy consumption, energy utilization, energy efficiency, and environmental impact, providing valuable references for further research of the Energy Industry and promoting high-quality development of the Energy Industry. Therefore, this article will draw on the existing research results and ideas, combined with China's national economic industry classification method, to include industries such as coal mining, petroleum and natural gas extraction, fuel processing, and electricity, heat, and gas production and supply into the scope of the Energy Industry so as to provide scientific support and an authoritative data resource guarantee for the collaborative evolution research conducted in this article.

### 2.3. The Digital Economy Core Industry and the Energy Industry in Terms of Synergistic Development Relationship

Scholars in relevant fields have conducted a series of explorations. Ozturk et al. studied the impact of digital finance inclusiveness on economic growth and environmental

sustainability in 42 One Belt and Road Initiative (OBRI) countries using the methods of pooled ordinary least squares (OLS), two-stage least squares (2SLS), and generalized method of moments (GMM), and they found that digital finance inclusiveness promoted economic growth through the surge in carbon dioxide emissions [24]. Ren et al. empirically analyzed the significant positive correlation between internet development and energy consumption scale and the significant negative correlation between internet development and energy consumption structure [25]. Hao et al. analyzed the direct and indirect impact mechanisms of internet development level on electricity intensity based on the sample data of 30 provinces in China, and found that internet development has a significant negative effect on electricity intensity, negative spatial spillover effect, and threshold effect [26]. Liu et al. studied the relationship between digital economic development, industrial structure upgrading, and global trade plans based on the pulse response function and intermediate effect model using data from 286 cities in China and found that the digital economy has long-term effects on promoting China's green total factor productivity (GTFP) [27]. Usman et al. adopted the boundary inspection method of the cointegration and error correction model to analyze the impact of information and communication technology (ICT) on the economic performance and energy consumption of South Asian economies and found that improving ICT technology will promote economic growth and help improve energy efficiency [28]. Cao et al. quantitatively analyzed the impact of digital finance on China's energy and environmental performance and found that digital finance has significantly improved China's energy and environmental performance [29]. Wang et al. empirically analyzed that internet technology indirectly promoted green economic growth by guiding industrial structure upgrading based on panel data from 269 prefecture-level cities in China [30]. Li et al. found that the digital economy significantly moderates carbon emissions based on an extended stochastic impact by regression on population, affluence, and technology (STIRPAT) model and panel data from 30 provinces in China [31]. Shahbaz et al. found that digital economic growth has effectively promoted the growth of renewable energy consumption and renewable energy generation structures based on panel data from 72 countries [32]. Li et al. conducted a nonlinear analysis combining the spatial DURBIN model (SDM) and the panel threshold model (PTM) and found that the green integration of the digital economy and traditional industries is of great significance for carbon emissions reduction [33]. Xue et al. found that the digital economy has promoted economic growth and optimized energy efficiency and industrial structure based on panel data from 30 provinces and 205 cities in China [34].

In summary, previous studies analyzed the role of the digital economy in promoting high-quality economic development from macro, regional, and industrial perspectives and found that the digital economy can effectively promote the transformation and upgrading of the Energy Industry and has significant spatial spillover effects. However, at present, few scholars have deeply analyzed the co-evolution laws of the Digital Economy Core Industry and the Energy Industry from a two-sector perspective. Therefore, this direction is expected to become a hot research area to promote the coordinated development of industries and achieve high-quality development goals in the context of digitalization.

## 3. Research Process, Indicator System, and Model Construction

### 3.1. Research Process

Based on the inter-provincial panel data of 30 provinces in China from 2011–2021, this article establishes a scientific and reasonable industrial development indicator system, adopts the analysis model based on DEA Malmquist and grey correlation analysis, and combines the entropy weighting method, the improved criteria importance through intercriteria correlation (CRITIC) method, and the coupling coordination degree model to obtain the indicators of the level of co-evolution of the Digital Economy Core Industry and the Energy Industry. The specific process is shown in Figure 1.

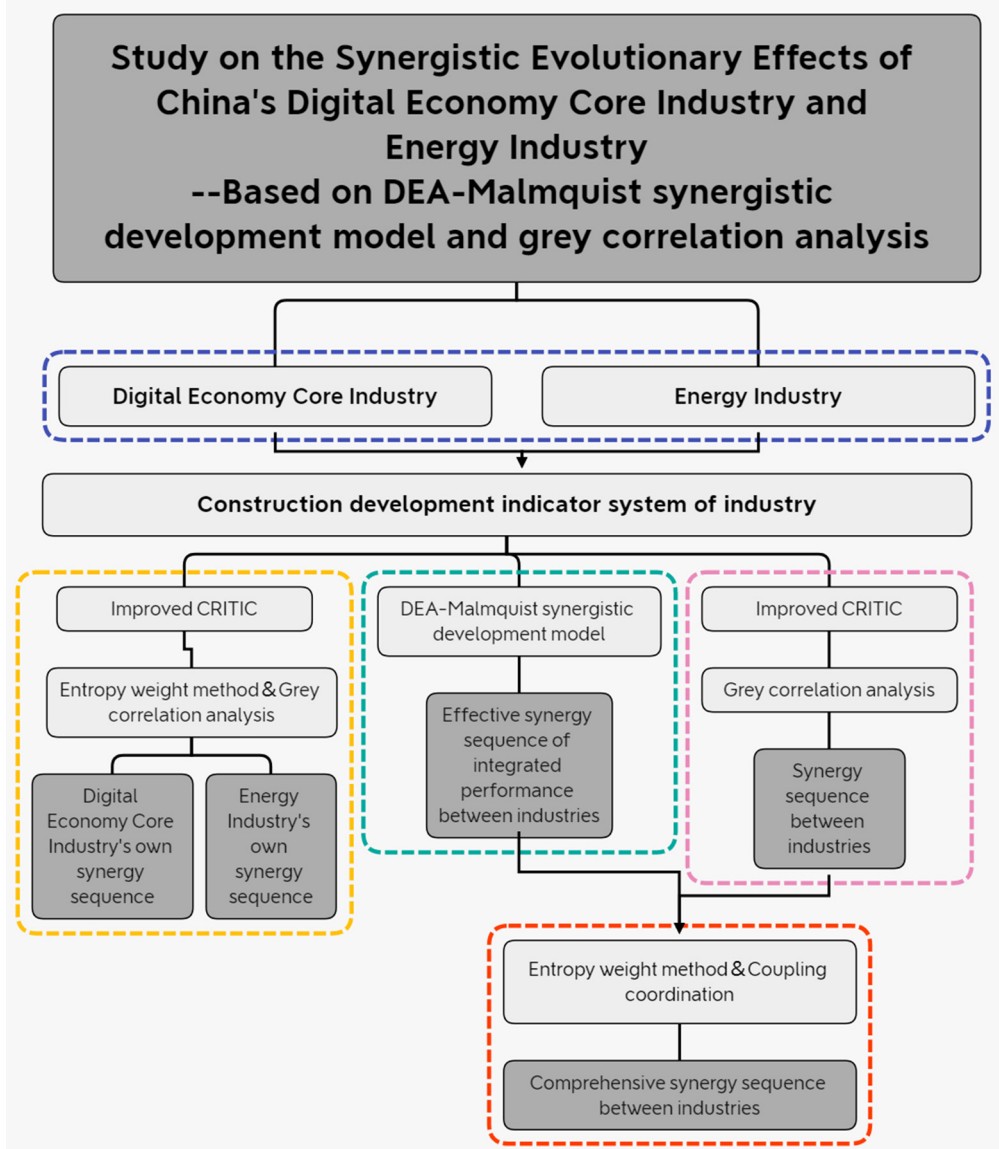

**Figure 1.** Flowchart of the research model on the synergistic evolution effect of China's Digital Economy Core Industry and Energy Industry.

*3.2. Indicator System*

3.2.1. Construction Development Indicator System of Digital Economy Core Industry

Currently, scholars frequently adopt indicators such as economic benefits, industry growth, social benefits, and growth potential to depict the characteristics of industrial development within an economy or specific industry. In this study, we have synthesized these indicator selection concepts and incorporated methodologies from existing research literature, both domestically and internationally, to formulate an industrial development indicator system. As a result, we have established 13 core industrial development indicators of the digital economy, distributed across four evaluative dimensions. These dimensions adhere to principles of scientific rigor, systematic coherence, and accessibility. The established indicators are presented in Table 1.

**Table 1.** Development indicator system of the Digital Economy Core Industry.

| Evaluation Dimension | Indicator Name | Indicator Definition | Unit | Positive and Negative Type | Input–Output Type |
|---|---|---|---|---|---|
| Economic Benefits | Digital Economy Scale | E-commerce sales + E-commerce procurement + Digital Economy Core Industry operating income | 100 Million Yuan | Positive | Output |
| | The profit margin of the main business | Total profit of Digital Economy Core Industry/Operating income of Digital Economy Core Industry | 100 Million Yuan | Positive | Output |
| | Fixed Asset Investment | Total fixed asset investment in Digital Economy Core Industry | 10 Thousand Yuan | Positive | Input |
| | Express business revenue as a percentage of total retail sales of social consumer goods | Express business revenue/Total retail sales of consumer goods | % | Positive | Output |
| Industry Development | Number of market entities | Number of enterprises in the Digital Economy Core Industry | Number | Positive | Input |
| | Number of employees | The number of employed persons in urban units in the Digital Economy Core Industry | 10 Thousand People | Positive | Input |
| | Industry per capita wage | Average Wages of Employed Persons in Urban Units in Digital Economy Core Industry | Yuan | Positive | Output |
| Social Benefits | Length of long-distance fiber optic cable lines | Long-distance fiber optic cable line length | 10 Thousand km | Positive | Input |
| | The average number of ports per Internet user | Internet broadband access ports/Internet broadband access users | Ports Per User | Positive | Input |
| | Percentage of e-commerce trading enterprises | The proportion of enterprises with e-commerce transaction activities | % | Positive | Output |
| Growth Potential | Technology Market Turnover | Technology Market Turnover | 100 Million Yuan | Positive | Output |
| | New product project development funds | Expenditure on the development of new products by industrial enterprises above the scale/number of new product projects by industrial enterprises above the scale | 10 Thousand Yuan Per Item | Positive | Input |
| | Local financial expenditure on science and technology | Local financial expenditure on science and technology | 100 Million Yuan | Positive | Input |

### 3.2.2. Construction Development Indicator System of the Energy Industry

Utilizing a method similar to the construction of a development indicator system for the Digital Economy Core Industry as aforementioned, 4 evaluation dimensions and 13 Energy Industry development indicators are constructed as shown in Table 2.

**Table 2.** Energy Industry development indicator system.

| Evaluation Dimension | Indicator Name | Indicator Definition | Unit | Positive and Negative Type | Input–Output Type |
|---|---|---|---|---|---|
| Economic Benefits | Energy Industry Scale | Energy Industry operating income | 10 Thousand Yuan | Positive | Output |
| | Energy Industry Investment | Energy Industry Investment | 100 Million Yuan | Positive | Input |
| | Fixed Asset Investment | Total fixed asset investment in Energy Industry | 10 Thousand Yuan | Positive | Input |
| | Per capita energy consumption | Energy consumption/year-end resident population | Tons of Standard Coal Per Person | Positive | Output |
| Industry Development | Number of market entities | Number of enterprises in the Energy Industry | Number | Positive | Input |
| | Number of employees | The number of employed persons in the urban units in Energy Industry | 10 Thousand People | Positive | Input |
| | Industry per capita wage | Average Wages of Employed Persons in Urban Units in the Energy Industry | Yuan | Positive | Output |
| Ecological Benefit | Discharge of chemical oxygen demand in wastewater | Discharge of chemical oxygen demand in wastewater | 10 Thousand Ton | Negative | Output |
| | Emission of sulfur dioxide in the exhaust gas | Emission of sulfur dioxide in the exhaust gas | 10 Thousand Ton | Negative | Output |
| | Investment completed for wastewater and exhaust gas treatment projects | Investment completed for wastewater treatment project + Investment completed for waste gas treatment project | 10 Thousand Yuan | Positive | Input |
| Growth Potential | Electricity consumption | Electricity consumption | 100 Million kWh | Positive | Output |
| | Number of invention patent applications | Number of invention patent applicationsby industrial enterprises above the scale | Piece | Positive | Input |
| | Local financial expenditure on resources exploration, electricity information and other affairs | Local financial expenditure on resources exploration, electricity information and other affairs | 100 Million Yuan | Positive | Input |

*3.3. Model Construction*

3.3.1. Research Model of Industry' Own Synergistic Evolutionary Effect

- Constructing a comprehensive sequence of the subsystem evaluation dimension based on the improved CRITIC method

In this article, we refer to the existing studies [35] and adopt the improved CRITIC method to construct a comprehensive series for each industrial development indicator in the evaluation dimension in response to the problems of the traditional CRITIC method.

First, assuming that the Digital Economy Core Industry and Energy Industry are two subsystems with synergistic development, the original indicator time series $x_{iqj}$ of j industrial development indicators in dimension q for subsystem i is constructed from the raw provincial panel data in year k and expressed as follows:

$$\mathbf{x_{iqj}} = \left(x_{iqj}(1), x_{iqj}(2), ...x_{iqj}(t)\right)^{T} \tag{1}$$

where $k = 1, 2, ...t$; $q = 1, 2, ...m$; $j = 1, 2, ...n$; then, $\mathbf{x_{iqj}(k)}$ is indicator **j** in dimension **q** for subsystem **i** in year **k**.

Next, the original data are separated with **max–min** standardization to eliminate dimensional effects. Then, the standardized indicator sequence $\mathbf{x_{iqj}'}$ is represented as follows:

$$\mathbf{x_{iqj}'} = \left(x_{iqj}'(1), x_{iqj}'(2), ...x_{iqj}'(t)\right)^{T} \tag{2}$$

Considering the existence of the value 0 after standardization of the indicator, from the perspective of research integrity, credibility, and scientificity, this article assigns a small

positive quantity to the value 0 and shifts the overall indicator value to the right, expressed as follows:

$$\text{Positive: } x_{iqj}{}'(k) = 0.001 + \frac{x_{iqj}(k) - \min\{x_{iqj}(k)\}}{\max\{x_{iqj}(k)\} - \min\{x_{iqj}(k)\}} \tag{3}$$

$$\text{Negative: } x_{iqj}{}'(k) = 0.001 + \frac{\max\{x_{iqj}(k)\} - x_{iqj}(k)}{\max\{x_{iqj}(k)\} - \min\{x_{iqj}(k)\}} \tag{4}$$

Subsequently, the improved CRITIC method is used to calculate the weights $\omega_{iqj}$ of each indicator under the **q** evaluation dimension and the comprehensive sequence $X_{iq}$ of **i** subsystem **q** evaluation dimensions is further obtained as follows:

$$\omega_{iqj} = \frac{C_{iqj}}{\sum\limits_{j=1}^{n} C_{iqj}} \tag{5}$$

$$C_{iqj} = \frac{\sigma_{iqj}}{X_{iqj}} \sum_{k=1}^{t} (1 - |r_{qj}|) \tag{6}$$

$$X_{iq} = \sum_{j=1}^{n} \omega_{iqj} \times x_{iqj}{}'(t) \tag{7}$$

where $\sigma_{iqj}$ and $X_{iqj}$ are the standard deviation and mean of indicator **j** in **q** dimensions, respectively, $C_{iqj}$ is the amount of information contained in indicator **j** in **q** dimensions, and $r_{qj}$ is the correlation coefficient among indicators in **q** dimensions.

- Constructing the industry's own synergy sequence-based on the entropy weight method and grey correlation analysis

First, assuming that the distance between any two evaluation dimensions within subsystem **i** is $d_{iuv}(k)$, the sequence of distances between each evaluation dimension, $d_{iuv}$, is as follows:

$$d_{iuv}(k) = |X_{iu}(k) - X_{iv}(k)| \tag{8}$$

$$\mathbf{d_{iuv}} = (d_{iuv}(1), d_{iuv}(2), ... d_{iuv}(t))^{T}, \tag{9}$$

where $\mathbf{u = 1, 2, ...m; v = 1, 2, ... m; u \neq v}$.

Second, assuming that the correlation coefficient between any 2 evaluation dimensions within subsystem **i** is $r_{iuv}(k)$, the calculation of the correlation coefficient between each evaluation dimension becomes as follows:

$$r_{iuv}(k) = \frac{\min\limits_{m}\min\limits_{t}|d_{iuv}(k)| + \rho\max\limits_{m}\max\limits_{t}|d_{iuv}(k)|}{|d_{iuv}(k)| + \rho\max\limits_{m}\max\limits_{t}|d_{iuv}(k)|} \tag{10}$$

where $\rho$ refers to the general treatment method, i.e., it takes the value of 0.5.

Subsequently, the entropy weight method is applied to calculate the distance weights $\omega_{iuv}$ among each evaluation dimension; then, the comprehensive synergy degree sequence $b_i(k)$ of the **i** subsystem's own indicators is expressed as follows:

$$\omega_{iuv} = \frac{1 - e_{iuv}}{m - \sum\limits_{u,v=1}^{m} e_{iuv}} \tag{11}$$

$$e_{iuv} = \frac{1}{\ln t} \sum_{k=1}^{t} f_{iuv} \ln f_{iuv} \tag{12}$$

$$f_{iuv} = r_{iuv}(k) \times \sum_{k=1}^{t} r_{iuv}(k) \tag{13}$$

$$b_i(k) = \sum_{u,v=1}^{m} \omega_{iuv} \times r_{iuv}(t) \tag{14}$$

where $e_{iuv}$ and $f_{iuv}$ are the entropy value and feature weight of the correlation coefficient between any two evaluation dimensions within subsystem $i$, respectively.

3.3.2. Research Model of Synergistic Evolutionary Effects between Industries

- Constructing a subsystem integrated sequence-based on the Improved CRITIC method

First, based on the previous model, the improved CRITIC method is used to calculate the weight $\omega_{iqj}'$ of each indicator within subsystem $i$. Further, the integrated sequence $X_i$ of subsystem $i$ is obtained as follows:

$$\omega_{iqj}' = \frac{C_{iqj}'}{\sum\limits_{q=1}^{m} C_{iqj}'} \tag{15}$$

$$C_{iqj}' = \frac{\sigma_{iqj}}{X_{iqj}} \sum_{k=1}^{t} \left(1 - \left| r_{qj}' \right| \right) \tag{16}$$

$$X_i = \sum_{q=1}^{m} \omega_{iqj}' \times x_{iqj}(t) \tag{17}$$

where $C_{iqj}'$ is the amount of information contained in each indicator within subsystem $i$, and $r_{qj}'$ is the correlation coefficient between indicators within subsystem $i$.

Second, assuming that the subsystem of the Digital Economy Core Industry is labeled **D**, the Energy Industry subsystem is labeled **S**, the integrated sequence of the subsystem of the Digital Economy Core Industry is $X_D$, and the integrated sequence of the Energy Industry subsystem is $X_S$, then the following obtain:

$$X_D = \sum_{q=1}^{m} \omega_{Dqj}' \times x_{Dqj}(t) \tag{18}$$

$$X_S = \sum_{q=1}^{m} \omega_{Sqj}' \times x_{Sqj}(t) \tag{19}$$

Among them, $\omega_{Dqj}'$ is the weight of each indicator within the subsystem of the Digital Economy Core Industry calculated through the improved CRITIC method, and $\omega_{Sqj}'$ is the weight of each indicator within the Energy Industry subsystem calculated through the improved CRITIC method.

- Constructing synergy sequences between industries based on grey correlation analysis

First, assuming that the distance between industrial subsystems is $d_{DS}(k)$, then the distance sequence $d_{DS}$ between subsystems is as follows:

$$d_{DS}(k) = |X_D(k) - X_S(k)| \tag{20}$$

$$d_{DS} = (d_{DS}(1), d_{DS}(2), \cdots d_{DS}(t))^T \tag{21}$$

Second, assuming that the correlation coefficients between industrial subsystems are $r_{DS}(k)$, the calculation for the correlation coefficients between industrial subsystems becomes as follows:

$$r_{DS}(k) = \frac{\min\limits_{m}\min\limits_{t}|d_{DS}(k)| + \rho\max\limits_{m}\max\limits_{t}|d_{DS}(k)|}{|d_{DS}(k)| + \rho\max\limits_{m}\max\limits_{t}|d_{DS}(k)|} \tag{22}$$

In addition, the synergy coefficient between two subsystems is a $1 \times \mathbf{k}$ matrix, so the synergy coefficient and correlation coefficients are the same between the two subsystems.

As a result, the synergy sequences $\mathbf{b_{DS}(k)}$ between industrial subsystems is represented as follows:

$$b_{DS}(k) = r_{DS}(k) \tag{23}$$

### 3.3.3. Research Model on the Effective Synergistic Evolutionary Effect of Integrated Performance between Industries-Based on the DEA Synergy Development Model

The Data Envelopment Analysis (DEA) model, a nonparametric method first introduced by Charnes et al. [36] (known as the CCR model), evaluates the relative efficiency of a Decision-Making Unit (DMU) when handling multiple inputs and outputs. However, its application is limited to efficiency assessment situations where DMUs exhibit constant returns to scale. To overcome this limitation, Banker et al. [37] extended the model to create the BCC model, which allows for the analysis of DMUs demonstrating variable returns to scale. This advancement further facilitated the derivation of pure technical efficiency and scale efficiency based on the original CCR model. Since these initial developments, subsequent scholars have continued to refine and expand upon these models in various forms.

The research object of this subsection is the effective synergy between two subsystems of the Digital Economy Core Industry and the Energy Industry from 2012 to 2021, and this article refers to the research method of He et al. [38] and expands it appropriately, using the input–output relationship to describe the synergy between two subsystems. At the same time, considering the lack of description of dynamic changes in traditional DEA methods, the absence of the premise of constant economies of scale in the actual development process of the industry, and the susceptibility of conclusions to special years or data combined with the characteristics of the data in this article, the DEA (BCC) Malmquist model is finally chosen to carry out the research.

For the Malmquist productivity indicator method, its decomposed expression form can be expressed as follows:

$$M_{(x_t, y_t, x_{t+1}, y_{y+1})} = \frac{S_t(x_t, y_t)}{S_t(x_{t+1}, y_{y+1})} \times \frac{D_t(x_{t+1}, y_{t+1})}{D_t(x_t, y_t)} \times \left[ \frac{D_t(x_{t+1}, y_{t+1})}{D_{t+1}(x_{t+1}, y_{t+1})} \times \frac{D_t(x_t, y_t)}{D_{t+1}(x_t, y_t)} \right]^{\frac{1}{2}}. \tag{24}$$

The first term on the right side of the equation indicates the change in scale efficiency (sech), whose value is greater than 1 indicates that the change in the scale of factor inputs related to industrial development makes the efficiency increase; the second term is the change in pure technical efficiency (pech), whose value is greater than 1 indicates that the change in the level of resource allocation and utilization related to industrial development makes the efficiency increase; the third term indicates the technology change (techch), whose value greater than 1 indicates that the change in the level of technology related to industrial development has led to technological progress.

At present, in the study of the integrated synergistic evolution effect between different systems, few studies include the effective synergistic effect of the integrated performance between systems as part of it and participate in the integrated synergistic evolution between systems. In this article, based on the DEA Malmquist method, we appropriately expand the model to include the scale efficiency change (sech), the pure technical efficiency change (pech), and the technological change (techch) as three indicators $\mathbf{e}$ describing the development of the integrated performance level among the target systems, which are denoted as inter-system development validity (marker $\mathbf{se}$), inter-system synergistic validity (marker $\mathbf{pe}$), and inter-system technical validity (marker $\mathbf{te}$), respectively, and the following assumptions are made based on the model as mentioned above.

First, assuming that the performance synergy sequence of the subsystem of the Digital Economy Core Industry to the Energy Industry subsystem is $\mathbf{E_{DSe}(k)}$, its descriptive indi-

cator $E_{DSe}$ has inter-system development validity as $E_{DSse}$, inter-system synergy validity as $E_{DSpe}$, and inter-system technical validity as $E_{DSte}$.

$$E_{DSse} = (E_{DSse}(1), E_{DSse}(2), ...E_{DSse}(t))^T \tag{25}$$

$$E_{DSpe} = (E_{DSpe}(1), E_{DSpe}(2), ...E_{DSpe}(t))^T \tag{26}$$

$$E_{DSte} = (E_{DSte}(1), E_{DSte}(2), ...E_{DSte}(t))^T \tag{27}$$

$$E_{DSe}(k) = \left( E_{DSse}(k), \ E_{DSpe}(k), E_{DSte}(k) \right) \tag{28}$$

Assuming that the performance synergy sequence of the Energy Industry subsystem to the subsystem of the Digital Economy Core Industry is $E_{SDe}(k)$, its descriptive indicator $E_{SDe}$ has inter-system development validity as $E_{SDse}$, inter-system synergy validity as $E_{SDpe}$, and inter-system technology validity as $E_{SDte}$.

$$E_{SDse} = (E_{SDse}(1), E_{SDse}(2), ...E_{SDse}(t))^T \tag{29}$$

$$E_{SDpe} = (E_{SDpe}(1), E_{SDpe}(2), ...E_{SDpe}(t))^T \tag{30}$$

$$E_{SDte} = (E_{SDte}(1), E_{SDte}(2), ...E_{SDte}(t))^T \tag{31}$$

$$E_{SDe}(k) = \left( E_{SDse}(k), \ E_{SDpe}(k), E_{SDte}(k) \right) \tag{32}$$

Second, the integrated performance synergy sequence of the subsystem of the Digital Economy Core Industry to the Energy Industry subsystem is assumed to be $M_{DSe}(k)$; the integrated performance synergy sequence of the Energy Industry subsystem to the subsystem of the Digital Economy Core Industry is assumed to be $M_{SDe}(k)$.

$$M_{DSe}(k) = E_{DSse}(k) \times E_{DSpe}(k) \times E_{DSte}(k) \tag{33}$$

$$M_{SDe}(k) = E_{SDse}(k) \times E_{SDpe}(k) \times E_{SDte}(k) \tag{34}$$

Subsequently, assuming that the integrated performance effective synergy sequence between the subsystem of the Digital Economy Core Industry and the Energy Industry subsystem is $Z_{DS}(k)$, the integrated performance effective synergy degree sequence between industries $Z_{DS}(k)$ can be expressed as follows:

$$Z_{DS}(k) = \frac{\min(M_{DSe}(k), M_{SDe}(k))}{\max(M_{DSe}(k), M_{SDe}(k))} \tag{35}$$

### 3.3.4. Research Model of Comprehensive Synergistic Evolutionary Effect between Industries Based on Entropy Weight Method and Coupling Coordination

First, from the model mentioned above, we can obtain the inter-industry subsystem synergy degree sequence as $b_{DS}(k)$ and the integrated performance effective synergy degree sequence among the industrial subsystems as $Z_{DS}(k)$, both of which are $1 \times k$ matrices.

Subsequently, it is assumed that the integrated synergy degree sequence between the subsystem of the Digital Economy Core Industry and the Energy Industry subsystem is defined as $B_{DS}(k)$, the integrated coordination indicator between industry subsystems is $T_{DS}(k)$, the coupling correlation degree between industry subsystems is $U_{DS}(k)$, the development coefficient of the synergy degree sequence $b_{DS}(k)$ between industry subsystems is $\alpha$, and the development coefficient of the integrated performance effective synergy degree sequence $Z_{DS}(k)$ between industry subsystems is $\beta$.

Then, the development coefficients of $\alpha$ and $\beta$ can be obtained by applying the entropy above weight method model, and the comprehensive coordination indicator $T_{DS}(k)$ among industrial subsystems can be expressed as follows according to the coupling coordination model:

$$T_{DS}(k) = \alpha \times b_{DS}(k) + \beta Z_{DS}(k) \tag{36}$$

The coupling correlation degree $U_{DS}(k)$ between industrial subsystems is expressed as follows:

$$U_{DS}(k) = \frac{\sqrt{b_{DS}(k) \times Z_{DS}(k)}}{(b_{DS}(k) + Z_{DS}(k))} \tag{37}$$

As a result, the comprehensive synergy sequence $B_{DS}(k)$ among industrial subsystems is expressed as follows:

$$B_{DS}(k) = \sqrt{T_{DS}(k) \times U_{DS}(k)} \tag{38}$$

## 4. Data Indicators, Empirical Analysis and Discussion of Results

### 4.1. Data Sources and Indicators Selection

#### 4.1.1. Data Sources

The data in this article are inter-provincial panel data of 30 provinces in China from 2011–2021, obtained from the China Statistical Yearbook, the China Statistical Yearbook of Information Industry, the China Statistical Yearbook of Science and Technology, the China Statistical Yearbook of Energy, the China Statistical Yearbook of Environment, and the annual statistical yearbooks or official websites of statistical bureaus in each province, municipality, and autonomous region. The missing data shall be supplemented by the linear interpolation method and the average value method according to the data of similar years.

#### 4.1.2. Indicators Selection

The research object of this article is the study of the comprehensive synergistic evolution between China's Digital Economy Core Industry and Energy Industry. For the selection of indicators, based on the reference of existing research results, the industrial development indicator system is constructed in accordance with the principles of scientificity, systematization, and accessibility by combining China's latest National Economic Classification of Industries (GB/T4754-2017).

For Digital Economy Core Industry, this article defines them as C (manufacturing industry: C39 (computer, communication and other electronic equipment manufacturing industry)), I (information transmission, software and information technology service industry: I63 (telecommunications, radio and television and satellite transmission services), I64 (Internet and related services), and I65 (software and information technology service industry)) and establishes four evaluation dimensions, i.e., economic efficiency, industry development, social benefits, and growth potential, a total of 13 industrial development indicators. Among them, considering that the National Bureau of Statistics of China no longer publishes the value added of subdivided industrial sectors since 2012, this article uses annual e-commerce sales, annual e-commerce procurement volume, and annual business income of the industry as indicators of the scale of the digital economy; takes into account the express business income as a percentage of total retail sales of social consumer goods, which can better reflect the level of the digital economy; and uses the number of market entities, the number of employees, and industry per capita, three indicators which are closely related to the industry are used to describe the development status. From the perspective that infrastructure will affect the development of the industry, the average number of ports per Internet user is included in the indicator system, and the indicator of local financial expenditure on science and technology is used to reflect the degree of government support.

For the Energy Industry, this article defines it as B (mining industry: B06 (coal mining and washing industry), B07 (oil and gas mining industry)), C (manufacturing industry: C25 (oil, coal and other fuel processing industry)), D (electricity, heat, gas and water production

and supply industry: D44 (electricity, heat production and supply industry), D45 (gas production and supply industry), D46 (water production and supply industry)), and establishes four evaluation dimensions, i.e., economic efficiency, industry development, ecological efficiency, and growth potential, with a total of 13 industrial development indicators. In order to ensure consistency with the indicators of the Digital Economy Core Industry, the annual operating income of the industry is used as the scale indicator of the Energy Industry; the per capita energy consumption is used to reflect the level of economic efficiency; from the perspective of consistency of indicators, the number of market entities, the number of employees and the per capita salary of the industry are used to describe the development status; and the chemical oxygen demand emissions in wastewater, sulfur dioxide emissions in exhaust gas, and the treatment of wastewater, sulfur dioxide, and sulfur dioxide emissions are used to describe the development status of the industry. Emissions of chemical oxygen demand in wastewater, sulfur dioxide emissions in waste gas, and the completed investment in wastewater and waste gas treatment projects are added to the system construction as indicators that can effectively reflect the ecological benefits of the industry, and the expenditure on local financial resources exploration, power information, and other affairs is included to reflect the degree of government support.

*4.2. Empirical Analysis*

4.2.1. Analysis of Industry's Own Synergistic Evolutionary Effect

Based on the research model of the synergistic evolutionary effect of industries themselves, the sequence of China's Digital Economy Core Industry and the Energy Industry's own synergy can be obtained for the period from 2012 to 2021 (as shown in Figure 2 and Table 3). Among them, the improved CRITIC method was used to calculate the weight coefficients of each indicator under each evaluation dimension (see in Appendix A, Tables A2 and A4), and the entropy weight method was used to calculate the distance weight coefficients between each evaluation dimension (see in Appendix A, Tables A3 and A5).

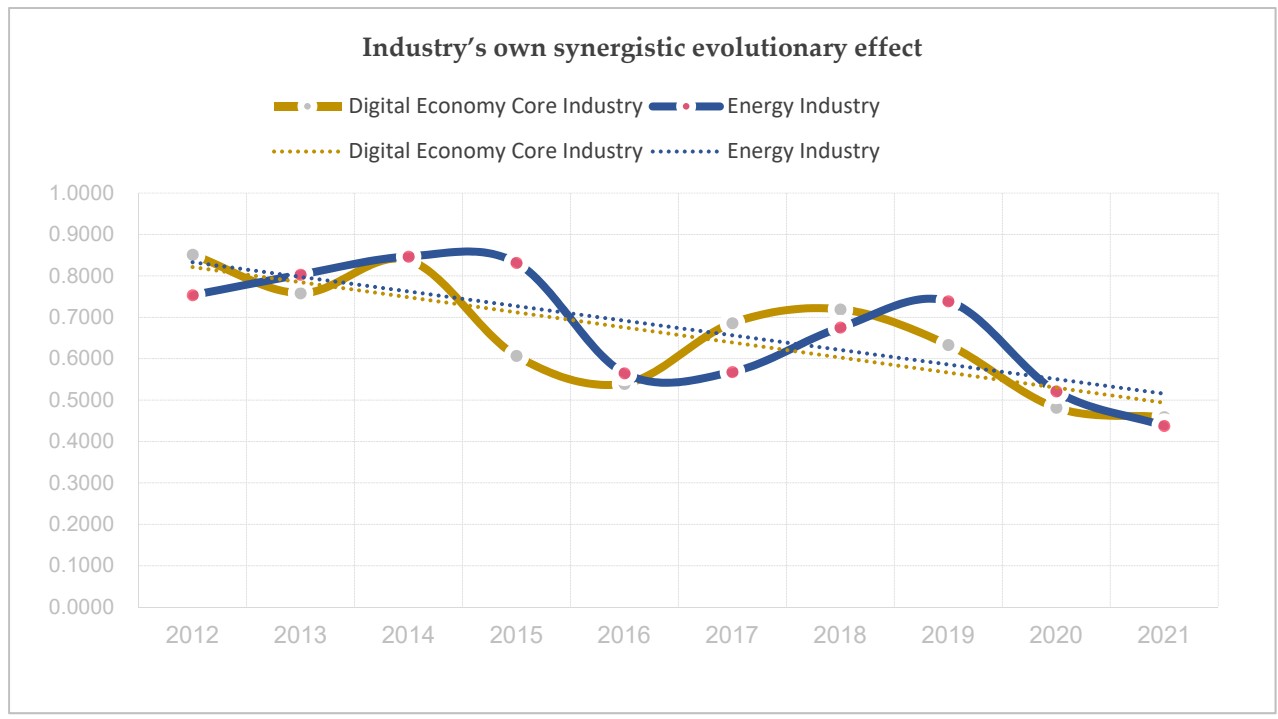

**Figure 2.** Trend in the synergistic evolutionary effects of the Digital Economy Core Industry and that of the Energy Industry itself, 2012–2021.

**Table 3.** Sequence of the synergy of China's Digital Economy Core Industry and that of the Energy Industry itself, 2012–2021.

| Year | 2012 | 2013 | 2014 | 2015 | 2016 |
|---|---|---|---|---|---|
| Digital Economy Core Industry ($b_D$) | 0.8512 | 0.7582 | 0.8382 | 0.607 | 0.5396 |
| Energy Industry ($b_S$) | 0.7535 | 0.8032 | 0.8469 | 0.8316 | 0.5650 |
| **Year** | **2017** | **2018** | **2019** | **2020** | **2021** |
| Digital Economy Core Industry ($b_D$) | 0.6861 | 0.7193 | 0.6333 | 0.4819 | 0.4590 |
| Energy Industry ($b_S$) | 0.5682 | 0.6755 | 0.7388 | 0.5212 | 0.4378 |

From the evolution of Table 3 and Figure 2, the following can be observed.

First, there is a more obvious consistency in the trend of the change of the two industries' own synergistic evolutionary effects from 2012–2021: the two own synergistic degree series has fluctuated down from 0.8512 and 0.7535 in 2012 to 0.4590 and 0.4378 in 2021, respectively.

Second, the two industries' own synergistic evolutionary effects had a short upward trend in 2013–2014 and 2016–2018, respectively, indicating that the industries' own synergistic evolutionary effects still function as the main driving force influencing the development of the industries at this stage.

Third, although the synergistic evolutionary effect of the industry itself plays a role as the main driving force, on the whole, the synergistic evolutionary effect of the two industries itself still shows an obvious decreasing trend, which initially indicates that the main driving force influencing the development of the industry is gradually shifting from the industry to other directions.

Fourth, from an overall perspective, the trend of the synergistic evolutionary effect of the two industries themselves fluctuates cyclically every five years from 2012 to 2021, which can be roughly divided into two fluctuation periods (2012–2016; 2017–2021), which can be interpreted as the potential impact of China's "Five-Year Plan" macro policy on industrial development. The potential impact on industrial development.

4.2.2. Analysis of the Synergistic Evolutionary Effect between Industries

Based on the construction of the inter-industry synergistic evolutionary effect research model, a sequence of synergy degrees between China's Digital Economy Core Industry and Energy Industry can be obtained for the period from 2012 to 2021, and the analysis results are shown in Tables 4 and 5 and Figure 3. Among them, the improved CRITIC method is used to calculate the weight coefficients of each indicator within the industry subsystem (see in Appendix A, Tables A6 and A7).

**Table 4.** Synergy sequence between China's Digital Economy Core Industry and Energy Industry, 2012–2021.

| Year | 2012 | 2013 | 2014 | 2015 | 2016 |
|---|---|---|---|---|---|
| Digital–Energy ($b_{DS}$) | 0.6440 | 0.5891 | 0.7806 | 0.3912 | 0.6524 |
| **Year** | **2017** | **2018** | **2019** | **2020** | **2021** |
| Digital–Energy ($b_{DS}$) | 0.7209 | 0.8654 | 0.5950 | 0.7630 | 1.0000 |

**Table 5.** The respective development sequence of China's Digital Economy Core Industry and Energy Industry, 2012–2021.

| Year | 2012 | 2013 | 2014 | 2015 | 2016 |
|---|---|---|---|---|---|
| Digital Economy Core Industry ($X_D$) | 0.0386 | 0.1404 | 0.2934 | 0.569 | 0.6487 |
| Energy Industry ($X_S$) | 0.1361 | 0.2581 | 0.3531 | 0.3316 | 0.5539 |
| **Year** | **2017** | **2018** | **2019** | **2020** | **2021** |
| Digital Economy Core Industry ($X_D$) | 0.6028 | 0.5500 | 0.5413 | 0.5386 | 0.5854 |
| Energy Industry ($X_S$) | 0.5283 | 0.5923 | 0.6567 | 0.4747 | 0.6060 |

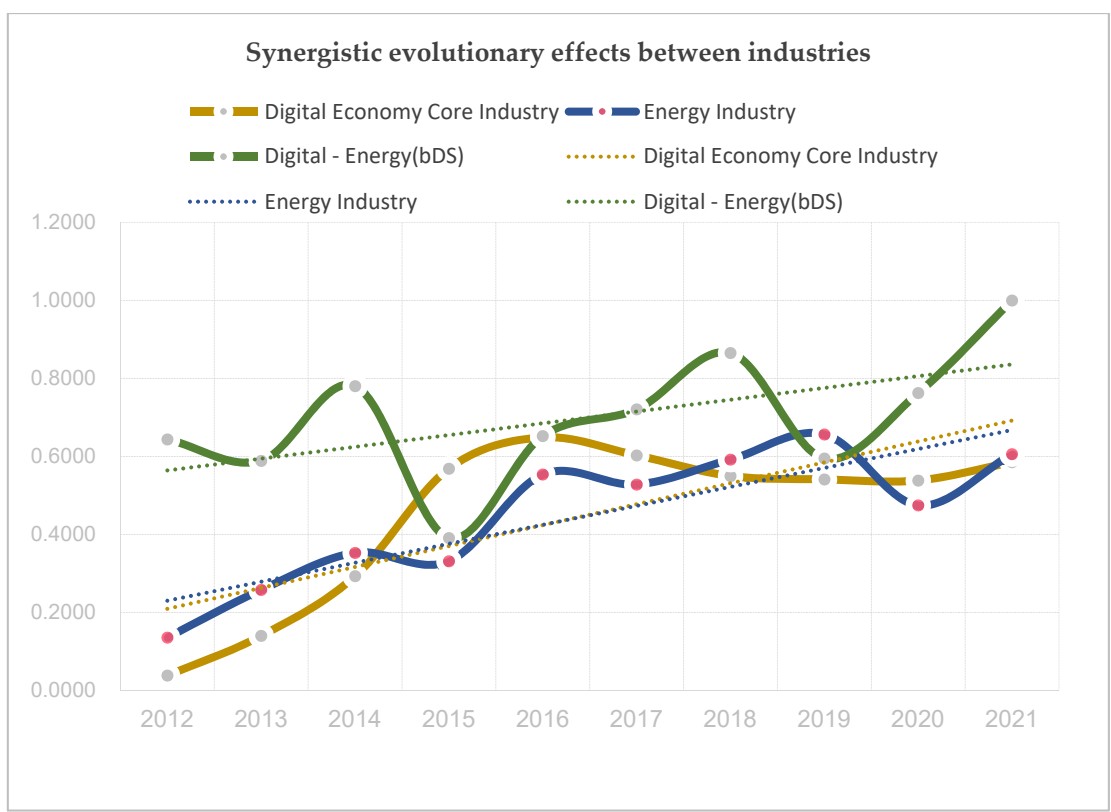

**Figure 3.** Trends in synergistic evolutionary effects between China's Digital Economy Core Industry and Energy Industry, 2012–2021.

From the evolution of Tables 4 and 5 and Figure 3, the following can be observed.

First, the synergy between the two industries from 2012–2021 shows a clear upward trend in the overall level, from 0.6440 in 2012 to 1.0000 in 2021.

Second, the trend of the synergistic evolutionary effect between the two industries fluctuates cyclically every five years, which can be roughly divided into two fluctuation cycles, 2012–2016 and 2017–2021, indicating that there is a potential influence relationship between the implementation of China's "Five-Year Plan" macro policy and industrial development. It can be found that the downward trend of the latter wave is slower than that of the former wave, and the upward trend of the latter wave is larger than that of the former wave.

Third, the synergistic evolutionary effect between the two industries had a short downward trend in 2014–2015 and 2018–2019, respectively, with an interval of about 5 years, further indicating the potential impact of macro policy making on industrial development. Further analysis shows that the decline in synergy between the two industries is due to the inconsistent direction of change in the development sequence of the two industries.

Fourth, combined with the characteristics above that both industries' own synergistic evolutionary effects show cyclical fluctuations and an overall decreasing trend, it can be preliminarily inferred that: the main driving force influencing industrial development has shifted from the industry's own synergistic effects to the inter-industry synergistic effects, and the inter-industry synergistic evolutionary effects bring new growth impetus to industrial development under the premise that the two industries' own synergistic evolutionary effects continue to decline.

4.2.3. Analysis of the Effective Synergistic Evolutionary Effect of Comprehensive Performance between Industries

Based on the research model of the effective synergistic evolutionary effect of integrated performance between industries, a sequence of an effective synergistic degree of

performance between China's Digital Economy Core Industry and Energy Industry can be obtained for the period from 2012 to 2021, and the analysis results are shown in Table 6 and Figure 4.

**Table 6.** Sequence of the effective synergy of integrated performance between China's Digital Economy Core Industry and Energy Industry, 2012–2021.

| Year | 2012 | 2013 | 2014 | 2015 | 2016 |
|---|---|---|---|---|---|
| Digital–Energy ($Z_{DS}$) | 0.8327 | 0.8753 | 0.7408 | 0.5293 | 0.8290 |
| **Year** | **2017** | **2018** | **2019** | **2020** | **2021** |
| Digital–Energy ($Z_{DS}$) | 0.9298 | 0.8293 | 0.7182 | 0.9194 | 0.7937 |

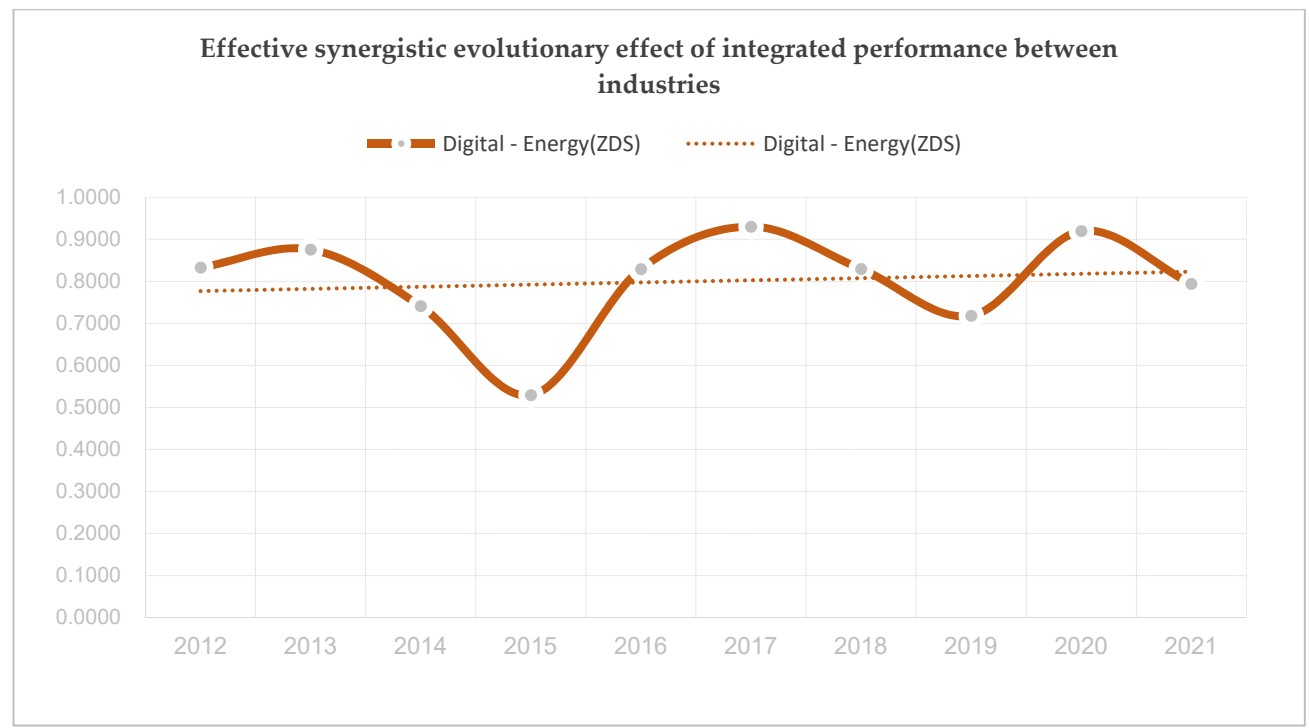

**Figure 4.** Trend of the effective synergistic evolutionary effect of integrated performance between China's Digital Economy Core Industry and Energy Industry, 2012–2021.

From the evolution of Table 6 and Figure 4, the following can be observed.

First, the series of the effective synergy of integrated performances between the two industries showed a fluctuating upward trend from 2012 to 2021, with the fluctuations gradually closing upward.

Second, the trend of the evolutionary effect of integrated performance effective synergy between the two industries fluctuates in a cycle of about every five years, which can be roughly divided into two fluctuation periods from 2012 to 2016 and from 2017 to 2021, and it can be found that the downward trend of the latter period is significantly slower than that of the former period when comparing the two periods before and after.

Thirdly, combining the aforementioned characteristics of the two industries' own synergistic evolutionary effects and inter-industry synergistic evolutionary effects, we can tentatively conclude that industrial development is potentially influenced by the implementation of China's Five-Year Plan and the endogenous driving force of industrial development has shifted from the industry's own synergistic effects to inter-industry synergistic effects. In the context of the declining synergistic evolution of the two industries,

inter-industry synergy is becoming a key influencing factor to improve the development performance of the Digital Economy and the Energy Industry.

4.2.4. Analysis of the Comprehensive Synergistic Evolutionary Effect between Industries

Based on the inter-industry comprehensive synergistic evolutionary effect research model, the effective synergy degree sequence of the performance of China's Digital Economy Core Industry and Energy Industry can be obtained from 2012 to 2021, and the analysis results are shown in Table 7 and Figure 5. The weight coefficients for the inter-industry synergistic evolutionary effect and the effective synergistic evolutionary effect of integrated performance between industries, determined using the entropy weight method, are provided in Table A8 of Appendix A.

**Table 7.** Sequence of the comprehensive synergistic evolutionary effect between China's Digital Economy Core Industry and Energy Industry, 2012–2021.

| Year | 2012 | 2013 | 2014 | 2015 | 2016 |
|---|---|---|---|---|---|
| Digital–Energy ($B_{DS}$) | 0.5879 | 0.5729 | 0.6202 | 0.4610 | 0.5903 |
| **Year** | **2017** | **2018** | **2019** | **2020** | **2021** |
| Digital–Energy ($B_{DS}$) | 0.6218 | 0.6539 | 0.5598 | 0.6338 | 0.6841 |

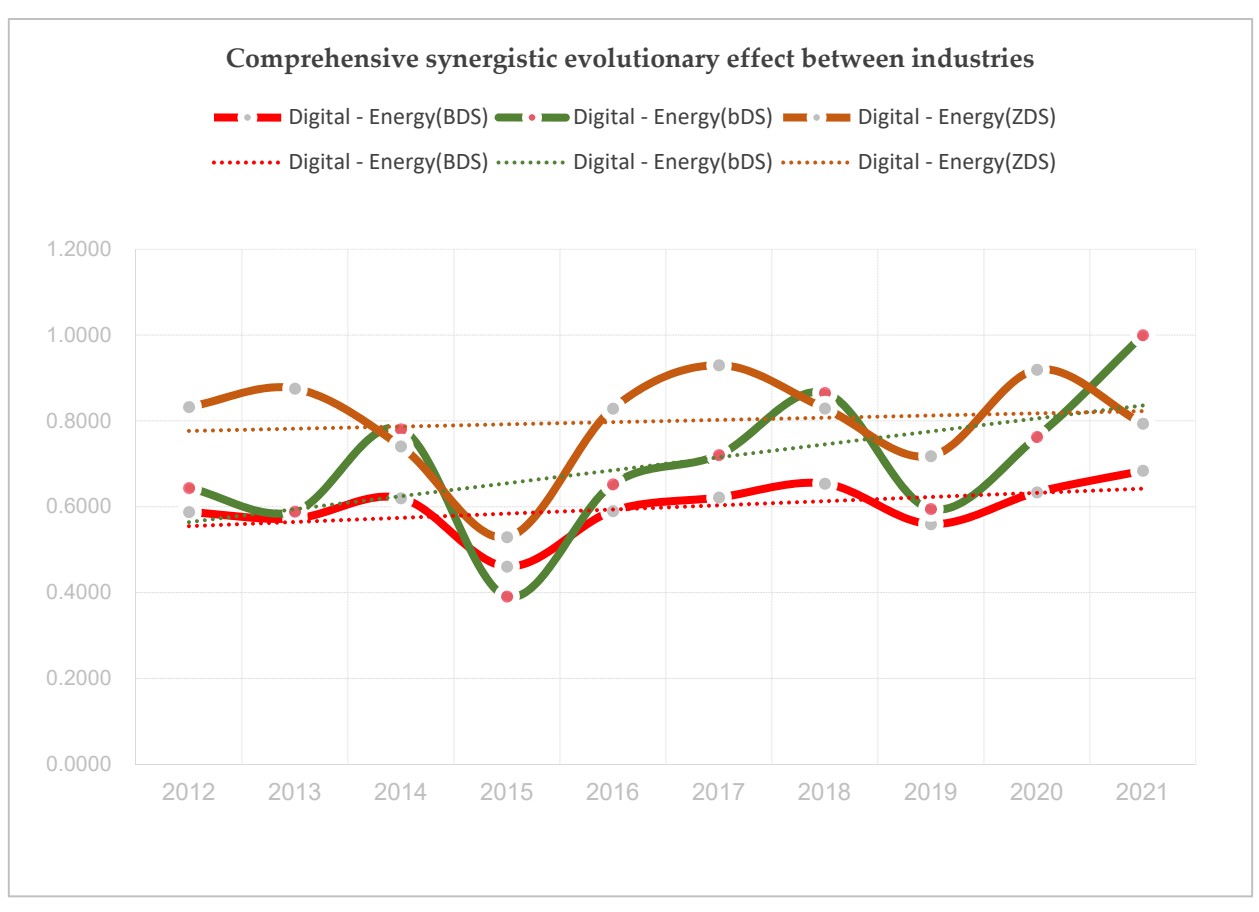

**Figure 5.** Trends in the integrated synergistic evolutionary effects between China's Digital Economy Core Industry and Energy Industry, 2012–2021.

From the evolution of Table 7 and Figure 5, the following can be observed.

First, the integrated synergy series between the two industries from 2012–2021 increased from 0.5879 in 2012 to 0.6841 in 2021, indicating a continuous increase in the synergy effect between them.

Second, the evolutionary trend of the comprehensive synergy between the two industries can be roughly divided into two fluctuation periods from 2012–2016 and 2017–2021, which fluctuate in a cycle of about every five years, further indicating the potential development of China's "Five-Year Plan" macro policy. The downward trend of the latter band has slowed down significantly compared to the former band, and the upward momentum of the latter band has increased significantly compared to the former band.

Third, the overall level of the integrated synergistic evolutionary effect between the two industries shows a clear upward trend, with short periods of decline in 2014–2015 and 2018–2019, respectively, with an interval of about 5 years, which further indicates the potential impact of macro policy formulation on industrial development.

Fourthly, combined with the aforementioned characteristics of the two industries' own synergistic evolutionary effect, the inter-industry synergistic evolutionary effect and the inter-industry comprehensive performance effective synergistic evolutionary effect, it can be further concluded that the main driving force affecting industrial development has been gradually transferred from the industry itself to the inter-industry, and under the premise that the two industries' own synergistic evolutionary effect continues to decline, the inter-industry comprehensive synergistic evolutionary effect can bring new growth momentum can be brought to industrial development.

*4.3. Results and Discussion*

This article examines the synergistic evolution effects between Digital Economy Core Industry and Energy Industry, with empirical results divided into two aspects: internal and external to the industries.

On the one hand, consistent and fluctuating decline in their own synergistic evolutionary effects from the internal view of the two industries, from 0.8512 and 0.7535 in 2012 to 0.4590 and 0.4378 in 2021, respectively, was noted. This trend indicates that the internal synergistic drivers influencing the development of both industries have followed a significant wave-like downward trend over the past decade. (1) This consistency might result from mutual influence and interdependence between the industries. (2) These synergistic evolutionary effects demonstrate two fluctuating waves, each with roughly a five-year cycle (2012–2016; 2017–2021), notably aligned with China's Five-Year Plan policy cycle. (3) During our study, the downward trend of the industries' own synergistic evolutionary effects was evident, signifying a shift in the driving force of industrial development from within the industries.

On the other hand, when viewed from the perspective of external effects, the synergistic evolution between the two industries increases from 0.6440 in 2012 to 1.0000 in 2021. Meanwhile, the effective synergistic evolution effects of comprehensive performance between industries fluctuates upwardly to 0.8000 from 2012 to 2021, and the comprehensive synergistic evolution effects between industries rises from 0.5879 in 2012 to 0.6841 in 2021. All three industries show a "weak downward and strong upward" trend of "two waves of fluctuation upward", which means that the synergistic evolutionary effect between the two industries externally shows a downward trend in individual periods but a significant wave upward trend in general. (1) The "weak downward and strong upward" trajectory could be attributed to the growing interdependence between the two industries. For instance, the core industries of the digital economy demand substantial electricity, while the Energy Industry relies on digital technology to enhance productivity and save costs. The intensifying interactions and dependencies are contributing to an increase in external synergistic evolutionary effects that influence industrial development. (2) The effects between the two industries also present two wave-like fluctuations with an approximate five-year cycle (2012–2016; 2017–2021), which correlates positively with China's "five-year plan" macro policy cycle. Consequently, enhancing cross-cycle macro policy alignment is crucial for

promoting the high-quality integrated development of both industries. (3) The external effects between the two industries have shown a significant upward fluctuation, affirming that these synergistic evolutionary effects have become a new growth engine for the development of the digital economy and Energy Industry.

In summary, the driving force for promoting the development of the Digital Economy Core Industry and Energy Industry in China has been shifted from internal to external factors. The traditional method of promoting industry development based on internal synergistic effects has shown signs of fatigue, while the external synergistic effects between industries have become a new driving force for promoting industrial development. This phenomenon is influenced by China's macro policies, such as "industrial restructuring" in the "five-year plan", and is also related to the industries themselves and the inter-industry reasons.

## 5. Policy Recommendations and Research Outlook

Based on the results discussed above, this article proposes the following policy recommendations and research outlook.

### 5.1. Policy Recommendations

(1) Given the fluctuating downward trend of the synergistic effects between the two industries, the following recommendations are proposed for the development of the two industries, respectively. Digital economy: Bolster the "learning-by-doing" effect within digital economy firms, advocate for network infrastructure development, and endorse public resources' open sharing. Amplify establishing a legal environment advocating market fairness, network security, scientific ethics, personal information safety, and rights protection. Promote the initiation of an integrity system. Emphasize pilot reforms fostering differentiated and diversified digital economy sectors across distinct regions, focusing on the "digital divide" and regional innovation disparities. Utilize digital technology to dismantle data flow barriers. Leverage economic spheres and city clusters to intensify the radiating and driving effects of key nodes in the digital economy sector. Employ data factors to advance "dual circulation" and nurture a high-quality "digital intelligence economy." Cultivate a diverse and synergistic environment for the digital enterprise, digital talent, and university research populations. Establish a shared platform integrating digital economy industries, academia, and research. Encourage universities to facilitate the transfer and delivery of digital economy patented technologies and talents to enterprises. Energy industry: Endeavor to expedite the development of emerging technologies and applications within critical sectors such as oil and gas, hydrogen energy, and nuclear power. Enhance efforts towards breakthroughs in state-of-the-art technologies within the green, low-carbon Energy Industry to facilitate optimizing and transforming the industry's structure, gradually phasing out obsolete production capacities. Within hydrogen energy, nuclear power, and other crucial sectors, work on expanding and extending the industrial chain, bolstering the links and filling the chain's gaps. Encourage cooperation and mutually beneficial partnerships among enterprises, universities, and research institutes. Capitalize on the driving role of advanced demonstration zones or pairing assistance to foster diverse technological exchanges and knowledge intersections, establishing an Energy Industry information exchange platform. Depending on the national energy base policy, hasten the labor and capital shift from the industry's upstream sector to the downstream sector and expedite the downstream industry chain's extension from primary to deep and fine chemical processing.

(2) Synergistic effects between the two industries: As the synergistic effects between the Digital Economy and Energy Industry become increasingly instrumental in fostering their development, it is advisable to conceive and enforce a national development strategy that addresses both. In fully accommodating the digital transformation requisites of the Energy Industry, swift remediation of digital infrastructure deficits in key areas

and segments is recommended. Orderly advancement of the seamless integration of innovation, industry, talent, and capital chains pertinent to both sectors is paramount. Encouraging research institutions, Internet corporations, and industry frontrunners to carry out digital transformation pilot demonstrations within segmented Energy Industry areas is also beneficial. The protection of intellectual property rights ought to be strengthened, and innovative technologies, such as blockchain, federated learning, secure multi-party computation, and data sandboxes, should be used to facilitate the orderly flow of energy data across domains, regions, and levels. The pivotal role of big data in shortening and eliminating cognitive gaps among Energy Industry chains and segmented industries cannot be overstated. This aids in effectively minimizing costs and thresholds for integrating and innovating the digital economy with the Energy Industry while simultaneously enhancing the scope, depth, and precision of their integration.

(3)  Given the policy cycle of the "Five-Year Plan" influences the synergistic effects between the Digital Economy Core Industry and the Energy Industry, it is advisable to enhance alignment with major national strategic frameworks such as "15 new types of digital economy", "Digital China", "Manufacturing Powerhouse", and "Cloud Computing and Big Data Empowering Intelligent Industry." This should be considered when devising a nationwide development strategy for integrating the digital economy and the real economy, with the primary aim of mitigating the impact of macro policy cycles on the synergistic effects between the two sectors. Thoroughly integrating the attributes of the Energy Industry's digital transformation and upgrading, the focus should be on key areas such as energy Internet construction, deep integration of I.T. and O.T. systems in the energy sector, and the development and utilization of energy big data. This guidance should foster sustainable digital transformation and upgrade within the Energy Industry, including enhancing the interconnectivity and communication of the energy Internet, reducing the data-energy consumption ratio of energy interconnection terminals, and increasing the operational efficiency of the on-site energy Internet. The energy data governance should be strengthened to improve energy data's quality and value density, support energy technology innovation, and enhance the Energy Industry's management level. Special attention should be given to breaking through the bottlenecks of key sub-industries within China's energy sector, such as new energy generation, power transmission, distribution, and sales. This includes effectively promoting the safe and orderly integration of I.T. and O.T. systems, utilizing big data and artificial intelligence technology to deeply mine the potential value of high-value density energy data, and encouraging intelligent data applications from production and manufacturing to design, supply chain, sales, and other links. The goal is to form a closed loop of energy data circulation and application based on enterprises.

*5.2. Research Outlook*

(1)  Expand the research sample. This study used data from 30 provinces in China for research, not encompassing the entire nation. Subsequent research could extend the sample size to enhance the scope and robustness of the study.

(2)  In-depth exploration of influencing factors. This study mainly focused on the relationship between the co-evolutionary effect between internal and external industries in relation to industrial development. Subsequent research could delve deeper into the primary factors influencing the synergistic evolution of both industries.

(3)  Strengthen interdisciplinary research. The exploration of synergistic evolution effects among industries transcends the domains of economics and management, encompassing various interdisciplinary realms, including engineering and environmental science. Future studies can intensify interdisciplinary engagement, thereby providing robust theoretical underpinning to realize sustainable, high-quality industrial growth.

**Author Contributions:** Conceptualization, G.X. and J.Z.; Methodology, G.X.; Software, G.X.; Formal analysis, C.L.; Data curation, J.Z.; Writing—original draft, G.X. and J.Z.; Writing—review & editing, C.L. and J.S.; Supervision, G.X. and C.L. All authors have read and agreed to the published version of the manuscript.

**Funding:** This research received no external funding.

**Institutional Review Board Statement:** Not applicable.

**Informed Consent Statement:** Not applicable.

**Data Availability Statement:** Publicly available datasets were analyzed in this study. This data can be found here: https://data.stats.gov.cn/easyquery.htm?cn=C01 (accessed on 29 June 2023).

**Conflicts of Interest:** The authors declare no conflict of interest.

## Appendix A

**Table A1.** Reference table of mathematical notations.

| Mathematical Notation | Definition |
|---|---|
| $x_{iqj}$ | The original indicator time series of j industrial development indicators in dimension q for subsystem i |
| $x_{iqj}(k)$ | Indicator j in dimension q for subsystem i in year k |
| $x_{iqj}'$ | The standardized indicator sequence |
| $x_{iqj}'(k)$ | The standardized indicator |
| $\omega_{iqj}$ | The weights of each indicator under the q evaluation dimension |
| $X_{iq}$ | The comprehensive sequence of i subsystem q evaluation dimensions |
| $C_{iqj}$ | The amount of information contained in indicator j in q dimensions |
| $\sigma_{iqj}$ | The standard deviation of indicator j in q dimensions |
| $X_{iqj}$ | The mean of indicator j in q dimensions |
| $r_{qj}$ | The correlation coefficient among indicators in q dimensions |
| $d_{iuv}$ | The sequence of distances between each evaluation dimension |
| $d_{iuv}(k)$ | The distance between any 2 evaluation dimensions within subsystem i |
| $r_{iuv}(k)$ | The correlation coefficient between any 2 evaluation dimensions within subsystem i |
| $\omega_{iuv}$ | The distance weights among each evaluation dimension |
| $b_i(k)$ | The comprehensive synergy degree sequence of the i subsystem's own indicators |
| $e_{iuv}$ | The entropy value of the correlation coefficient between any two evaluation dimensions within subsystem i |
| $f_{iuv}$ | The feature weight of the correlation coefficient between any two evaluation dimensions within subsystem i |
| $\omega_{iqj}'$ | The weights of each indicator within subsystem i |
| $C_{iqj}'$ | The amount of information contained in each indicator within subsystem i |
| $r_{qj}'$ | The correlation coefficient between indicators within subsystem i |
| $X_i$ | The integrated sequence of subsystem i |
| $D$ | The subsystem of the Digital Economy Core Industry |
| $S$ | The Energy Industry subsystem |
| $X_D$ | The integrated sequence of the subsystem of the Digital Economy Core Industry |
| $X_S$ | The integrated sequence of the Energy Industry subsystem |
| $\omega_{Dqj}'$ | The weight of each indicator within the subsystem of the Digital Economy Core Industry |
| $\omega_{Sqj}'$ | The weight of each indicator within the Energy Industry subsystem |
| $d_{DS}$ | The distance sequence between subsystems |
| $d_{DS}(k)$ | The distance between industrial subsystems |
| $r_{DS}(k)$ | The correlation coefficients between industrial subsystems |
| $b_{DS}(k)$ | The synergy sequences between industrial subsystems |
| $E_{DSe}(k)$ | The performance synergy sequence of the subsystem of the Digital Economy Core Industry to the Energy Industry subsystem |
| $E_{DSse}$ | Inter-system development validity of the subsystem of the Digital Economy Core Industry to the Energy Industry subsystem |

**Table A1.** *Cont.*

| Mathematical Notation | Definition |
|---|---|
| $E_{DSpe}$ | Inter-system synergy validity of the subsystem of the Digital Economy Core Industry to the Energy Industry subsystem |
| $E_{DSte}$ | Inter-system technical validity of the subsystem of the Digital Economy Core Industry to the Energy Industry subsystem |
| $E_{SDe}(k)$ | The performance synergy sequence of the Energy Industry subsystem to the subsystem of the Digital Economy Core Industry |
| $E_{SDse}$ | Inter-system development validity of the Energy Industry subsystem to the subsystem of the Digital Economy Core Industry |
| $E_{SDpe}$ | Inter-system synergy validity of the Energy Industry subsystem to the subsystem of the Digital Economy Core Industry |
| $E_{SDte}$ | Inter-system technology validity of the Energy Industry subsystem to the subsystem of the Digital Economy Core Industry |
| $M_{DSe}(k)$ | The integrated performance synergy sequence of the subsystem of the Digital Economy Core Industry to the Energy Industry subsystem |
| $M_{SDe}(k)$ | The integrated performance synergy sequence of the Energy Industry subsystem to the subsystem of the Digital Economy Core Industry |
| $Z_{DS}(k)$ | The integrated performance effective synergy sequence between the subsystem of the Digital Economy Core Industry and the Energy Industry subsystem |
| $B_{DS}(k)$ | The integrated synergy degree sequence between the subsystem of the Digital Economy Core Industry and the Energy Industry subsystem |
| $T_{DS}(k)$ | The integrated coordination indicator between industry subsystems |
| $U_{DS}(k)$ | The coupling correlation degree between industry subsystems |
| A | The development coefficient of the synergy degree sequence $b_{DS}(k)$ between industry subsystems |
| B | The development coefficient of the integrated performance effective synergy degree sequence $Z_{DS}(k)$ between industry subsystems |

**Table A2.** Section 4.2.1, China's Digital Economy Core Industry—improved CRITIC method for calculating weights.

| Industry | Evaluation Dimension | Indicator | Improved CRITIC Method |
|---|---|---|---|
| China's Digital Economy Core Industry | Economic Benefits | Digital economy scale | 20% |
| | | The profit margin of the main business | 42% |
| | | Fixed asset investment | 18% |
| | Industry Development | Express business revenue as a percentage of total retail sales of social consumer goods | 21% |
| | | Number of market entities | 32% |
| | | Number of employees | 26% |
| | | Industry per capita wage | 42% |
| | Social Benefits | Length of long-distance fiber optic cable lines | 32% |
| | | The average number of ports per Internet user | 51% |
| | | Percentage of e-commerce trading enterprises | 17% |
| | Growth Potential | Technology market turnover | 27% |
| | | Expenditure on the development of new products | 47% |
| | | Local financial expenditure on science and technology | 26% |

**Table A3.** Section 4.2.1 China's Digital Economy Core Industry—entropy weight method for calculating weights.

| | Indicator a–Indicator b | Entropy Weight Method |
|---|---|---|
| The synergistic evolutionary effects of the China's Digital Economy Core Industry itself | Economic Benefits–Industry Development | 7% |
| | Economic Benefits–Social Benefits | 20% |
| | Economic Benefits–Growth Potential | 18% |
| | Industry Development–Social Benefits | 22% |
| | Industry Development–Growth Potential | 21% |
| | Social Benefits–Growth Potential | 12% |

**Table A4.** Section 4.2.1 Energy Industry—improved CRITIC method for calculating weights.

| Industry | Evaluation Dimension | Indictor | Improved CRITIC Method |
|---|---|---|---|
| Energy Industry | Economic Benefits | Energy Industry scale | 36% |
| | | Energy Industry investment | 19% |
| | | Fixed asset investment | 16% |
| | | Energy consumption per capita | 29% |
| | Industry Development | Number of market entities | 27% |
| | | Number of employees | 35% |
| | | Industry per capita wage | 39% |
| | Ecological Benefits | Discharge of chemical oxygen demand in wastewater | 50% |
| | | Emission of sulfur dioxide in the exhaust gas | 20% |
| | | Investment completed for waste water and gas treatment project | 30% |
| | Growth Potential | Electricity consumption | 27% |
| | | Number of invention patent applications | 26% |
| | | Local financial expenditure on resources exploration, electricity information and other affairs | 47% |

**Table A5.** Section 4.2.1 Energy Industry—entropy weight method for calculating weights.

| | Indicator a–Indicator b | Entropy Weight Method |
|---|---|---|
| The synergistic evolutionary effects of the Energy Industry itself | Economic Benefits–Industry Development | 8% |
| | Economic Benefits–Ecological Benefits | 38% |
| | Economic Benefits–Growth Potential | 6% |
| | Industry Development–Ecological Benefits | 14% |
| | Industry Development–Growth Potential | 13% |
| | Ecological Benefits–Growth Potential | 21% |

**Table A6.** Section 4.2.2 China's Digital Economy Core Industry—improved CRITIC method for calculating weights.

| Industry | Indicator | Improved CRITIC Method |
|---|---|---|
| China's Digital Economy Core Industry | Digital economy scale | 5% |
| | The profit margin of the main business | 7% |
| | Fixed asset investment | 5% |
| | Express business revenue as a percentage of total retail sales of social consumer goods | 5% |
| | Number of market entities | 5% |
| | Number of employees | 4% |
| | Industry per capita wage | 5% |
| | Length of long-distance fiber optic cable lines | 5% |
| | The average number of ports per Internet user | 23% |
| | Percentage of e-commerce trading enterprises | 5% |
| | Technology market turnover | 7% |
| | Expenditure on the development of new products | 20% |
| | Local financial expenditure on science and technology | 5% |

**Table A7.** Section 4.2.2 Energy Industry—improved CRITIC method for calculating weights.

| Industry | Indicator | Improved CRITIC Method |
|---|---|---|
| Energy Industry | Energy Industry scale | 11% |
| | Energy Industry investment | 5% |
| | Fixed asset investment | 4% |
| | Energy consumption per capita | 8% |
| | Number of market entities | 4% |
| | Number of employees | 6% |
| | Industry per capita wage | 5% |
| | Discharge of chemical oxygen demand in wastewater | 24% |
| | Emission of sulfur dioxide in the exhaust gas | 6% |
| | Investment completed for waste water and gas treatment project | 9% |
| | Electricity consumption | 4% |
| | Number of invention patent applications | 4% |
| | Local financial expenditure on resources exploration, electricity information and other affairs | 8% |

**Table A8.** Section 4.2.4 The inter-industry synergistic evolutionary effect—entropy weight method for calculating weights.

| | The Synergistic Evolutionary Effect | Entropy Weight Method |
|---|---|---|
| The comprehensive synergistic evolutionary effect between industries | The inter-industry synergistic evolutionary effect | 72% |
| | The effective synergistic evolutionary effect of integrated performance between industries | 28% |

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
