# Peer review of "Study on the Synergistic Evolutionary Effects of China’s Digital Economy Core Industry and Energy Industry Based on DEA Malmquist Synergistic Development Model and Grey Correlation Analysis"

_sustainability, doi:10.3390/su151310382_

Round 1

Reviewer 1 Report

1. It seems that the mathematical notations are not explained in general. Please explain all mathematical notations used. 

2. Why do the authors switch between matrix and (summation) scalar notations? This is very confusing and the authors should stick to one.

3. Wouldn't there be more relevant industries than the energy industry for the digital economy? i.e. the IT industry, mobile communication industry, etc.

4. Were the variables deflated?

5. Shouldn't there be some control for spatial effects?

6. Please provide additional justifications/citations on the method used. 

Moderate changes are required for grammar and expression. It is recommended that the authors consider professional editing service.

For example (page 8, lines 245-247):

(as-is)

Similar to the construction method of development indicator system for the Digital Economy Core Industry mentioned above, 4 evaluation dimensions and 13 Energy Industry development indicators are constructed, as shown in Table 3.2.

(should be)

Utilizing a method similar to the construction of a development indicator system for the Digital Economy Core Industry as aforementioned, 4 evaluation dimensions and 13 Energy Industry development indicators are constructed as shown in Table 3.2.

Reviewer 2 Report

1.   English abbreviations (such as SGDI, CE, OECD, KPWW, SBM, OBRI, STIRPAT) are widely used in the literature review in the second section. It is recommended to use the full name for readers to read more clearly.

2.   The arrow below the blue dashed line in Figure 3.1 is wrong, please correct it.

3.     Pay attention to the rules for writing references.

4. The abscissa in Figure 4.1~Figure 4.4, only Figure 4.3 uses 2011~2020, which is quite inconsistent! Moreover, the data quoted in Figure 4.1 and Figure 4.2 are from 2012 to 2021; the data quoted in Figure 4.3 and Figure 4.4 are from 2011 to 2021, which is quite inconsistent!

5.  Whether there is an upward or downward trend in the curves in Figure 4.1~Figure 4.4, it is recommended to use a linear regression line to determine more accurately!

6.   The theoretical model of this paper is quite complete, but it is a pity that the results of quantitative data cannot be summarized in the conclusion!

Reviewer 3 Report

Please refer to the attached review report.

Round 2

Reviewer 1 Report

The reviewer would like to commend the efforts made by the authors in revising the manuscript. The manuscript quality has significantly improved. 

Although the application of spatial analysis can wait for future research, the issue of using deflated data seems to be quite important. Therefore, the reviewer would like to ask the authors to further validate the results with deflated data.    
